# An Overview of Opioid Prescription Patterns among Non-Opioid Users Following Emergency Department Admission

**DOI:** 10.3390/healthcare12111138

**Published:** 2024-06-03

**Authors:** Miriam Zeino, Romain Léguillon, Pauline Brevet, Baptiste Gerard, Catherine Chenailler, Johanna Raymond, Lucas Bibaut, Sophie Pouplin, Luc Marie Joly, Rémi Varin, Eric Barat

**Affiliations:** 1Department of Pharmacy, Rouen University Hospital, F-76000 Rouen, France; miriam.zeino@chu-rouen.fr (M.Z.); romain.leguillon@chu-rouen.fr (R.L.); catherine.chenailler@chu-rouen.fr (C.C.); johanna.raymond@chu-rouen.fr (J.R.); remi.varin@chu-rouen.fr (R.V.); 2Laboratoire d’Informatique Médicale et d’Ingénierie des Connaissances en e-Santé (LIMICS), U1142, IN-9 SERM, Sorbonne Université, F-75005 Paris, France; 3CIC-CRB 1404, Department of Rheumatology, CHU Rouen, University Rouen Normandie, F-76000 Rouen, France; pauline.brevet@chu-rouen.fr; 4Department of Rheumatology, Rouen University Hospital, F-76000 Rouen, France; baptiste.gerard@chu-rouen.fr (B.G.); lucas.bibaut@chu-rouen.fr (L.B.); sophie.pouplin@chu-rouen.fr (S.P.); 5Emergency Medicine Department, Universitary Hospital of Rouen, University Rouen Normandy, F-76000 Rouen, France; luc-marie.joly@chu-rouen.fr; 6Department of Public Health, Normandie University UNICAEN, Inserm U1086, F-14000 Caen, France

**Keywords:** opioid, pain management, emergency

## Abstract

The evolving landscape of opioid prescription practices necessitates a comprehensive understanding of emerging patterns, particularly among new opioid users discharged from emergency departments. This study delves into the intricate realm of opioid utilization by elucidating the prevalence of their prescriptions. A retrospective analysis of electronic health records was conducted, including a cohort of 71 patients who received opioid prescriptions upon discharge from emergency departments from 1 January 2022 to 30 June 2022. Demographic characteristics and prescription details were systematically examined. This study illuminates tramadol’s prominence, with 84% of prescriptions and a Defined Daily Dose (DDD) morphine equivalent of 60 mg, as the primary choice as a new opioid, a finding that draws attention due to the closely aligned dosages with morphine equivalents. This discovery prompts a critical reassessment of tramadol’s therapeutic role, considering its multifaceted nature encompassing serotonergic effects and heightened fall risks. This study advocates for a nuanced and vigilant approach to tramadol prescription, cognizant of its potential risks and therapeutic implications, and highlights the imperative of optimizing data quality and traceability within electronic health records to enhance patient care and facilitate future research endeavors.

## 1. Introduction

In recent decades, there has been a significant increase in the prescription of analgesics, as demonstrated in multiple studies to be the result of caring for non-cancer pain [1,2,3,4].

Even though this increase in opioid use is partly associated with the management of chronic pain [5], the inappropriate use of opioids for acute pain can lead to chronic opioid use [6]. One of the most striking examples is the use of opioids for dental pain [7].

The convergence of these factors emphasizes the critical need for tailored management and close monitoring to mitigate the risks associated with opioid use in the treatment of acute pain.

Despite the opioid overdose epidemic observed in the United States and Canada, there is a noticeable lack of nationwide European studies addressing this topical issue. In France, a different pattern of opioid consumption has been observed compared to its European neighbors, as highlighted by Hider-Mynarz et al. [8]. Their study, which analyzed prescription opioid sales trends from 2006 to 2015, revealed distinct patterns across Europe, with significantly higher opioid prescription use in countries like the United Kingdom. However, despite this difference in consumption patterns, it is crucial to note that there has also been a worrisome increase in opioid use in France, as demonstrated by the research conducted by [9]. These studies underscore a concerning rise in prescription opioid use, which is associated with a noteworthy increase in opioid-related morbidity and mortality in the country. This situation emphasizes the urgent need for proper monitoring and management of opioid consumption in France to address this growing issue.

In response to the emergence of health crises in certain countries and to ensure access to patients requiring opioid treatment, the French National Authority for Health (Haute Autorité de Santé) issued new guidelines in March 2022 for the management of acute pain [10]. The prescription of opioids is justified as a first-line treatment for severe acute pain (Numerical Rating Scale, NRS ≥ 6/10) when there is no etiological treatment available for rapid pain relief. They advise using only immediate-release formulations (no extended-release) for a maximum duration of 14 days. Furthermore, the guidelines for appropriate prescribing (not specific to opioids) emphasize specifying the International Non-proprietary Name (INN), dosage, and frequency of administration, along with indicating the prescribed duration. The duration of opioid prescription should be as short as possible and less than 14 days, with follow-up provided by the treating physician after discharge. Pain is the most common reason that patients present to the emergency department [11]. This has resulted in a significant increase in opioid analgesic prescriptions from emergency departments during the last decade, warranting special attention [12].

Our main objective is to thoroughly study opioid prescription patterns in patients following their visit to emergency department units. We aim to closely examine, the prescription, usage, and duration of opioids in these patients in a real-life study of professional practice to gain a better understanding of the trends and implications of this practice.

## 2. Materials and Methods

### 2.1. Patient Selection

We conducted an observational and retrospective study in a single center at the University Hospital of Rouen, covering the period from 1 January 2022 to 30 June 2022. This study (E2021-90) was approved by the Institutional Review Board (IRB) of a university hospital in France. Written patient consent was waived by the IRB.

Patient identification was based on the utilization of a health data warehouse called EdSAN (Health Data Warehouse of Normandy). We selected commercially available opioids in France, which were used as the query: fentanyl, hydromorphone, morphine, oxycodone, oxycodone + naloxone, pethidine, sufentanil, tapentadol, tramadol, tramadol + paracetamol, codeine, codeine + paracetamol, opium + paracetamol.

Patient characteristics, reason for hospitalization, prescribed opioid molecule(s), and prescription duration were recorded from each record and categorized. The inclusion criteria were as follows: adult patients experiencing acute pain, not currently receiving ongoing opioid treatment, and being discharged to their home after their emergency department visit. Only discharged patients were included because of the risk of misuse without medical supervision. Exclusion criteria were as follows: cancer-related pain, chronic pain, known history of opioid treatment, current opioid treatment, discharge without returning home, and those who experienced death during hospitalization. These carefully defined criteria aimed to ensure a homogeneous study population and to facilitate a meaningful interpretation of the results.

Additionally, this study utilized the concept of incidence to evaluate the occurrence rate of novel opioid users within this same population.

### 2.2. Definition of a Compliant Prescription

The guidelines of the French National Authority for Health were assessed to ensure prescription compliance. This comparison allowed us to assess the adherence of opioid prescriptions to the established HAS guidelines and to identify any potential variations or non-compliant practices.

### 2.3. Morphine Equivalent

We followed both national and international guidelines for establishing morphine dose equivalences to ensure accurate and standardized assessments [13,14] that align seamlessly with our established protocols and have undergone rigorous validation, as outlined in Table 1 for easy reference.

## 3. Results

### 3.1. Patient Selection

During the studied period, 2198 patients were hospitalized in the emergency department. Among them, 1112 were identified through the health data repository as potentially having received opioids based on our query. Furthermore, 71 patients were deemed eligible for inclusion (Figure 1). Deaths represent the patients who had received intravenous opioids for sedation at end-of-life.

In the cohort of individuals who utilized emergency services throughout the duration of this study, the prevalence of opioid consumption was approximated to be 13.7% before emergency and 16.9% after emergency. Around 3.2% of patients admitted to the emergency department were recognized as newly initiated consumers of opioids.

### 3.2. Patient Characteristics

The general characteristics of the population are available in Table 2. Among the 71 patients who received opioids in the emergency department, the mean age was 48.6 (±21.4) years, predominantly male (*n* = 40, 56.3%). The 45–65-years age group was the most represented, and the main reasons for seeking emergency care were either trauma or rheumatological issues. Regarding the assessment of pain upon admission to the emergency department, the mean score was 6.9 (±2.2) according to the NRS for pain. Data regarding the NRS outcomes were available only for 58 patients (81.7%). Among them, a majority (*n* = 39, 55%) had an NRS score ≥ 6.

### 3.3. Compliance of Opioid Prescriptions at Emergency Department Discharge

Among the 71 included patients, only 11 (16.6%) had a justified and compliant prescription of opioids, according to the French National recommendations [10].

For 20 patients (28.2%), an opioid prescription was issued despite an NRS < 6. In 12 cases (16.9%), an extended-release formulation was chosen as the initial approach, and for 14 patients (19.7%), the opioid prescription exceeded 14 days. The most frequently prescribed opioid was tramadol (*n* = 60, 84.5%), including only 10 (16.7%) compliant prescriptions. Details of the prescriptions are shown in Table 3.

### 3.4. Opioid Exposure at Emergency Department Discharge

Among the opioids, tramadol and morphine showed the highest cumulative and daily doses in morphine equivalent. The median dose of tramadol was 2000 mg as a cumulative dose, 400 mg as a morphine equivalent, 200 mg as a daily dose, and 40 mg as a morphine equivalent. In comparison, the cumulative dose of morphine was 350 mg, and 20 mg for daily dose. No significant difference was observed between the cumulative doses of tramadol and morphine (*p* = 0.85) or their daily doses (*p* = 0.22). The results are presented in Table 4.

## 4. Discussion

### 4.1. Population

One observation within our investigation pertains to the prevalence of opioid use within our study cohort, which stood at 13.7%. This figure demonstrates a remarkable proximity to the prevalence previously documented in a prior study [9].

An innovative aspect of our research lies in our endeavor to assess the incidence of new opioid users following their emergency department visit, a measure which, to our knowledge, has not been documented in the existing literature. Our findings revealed an incidence rate of new opioid consumers amounting to 3.2%. This observation, albeit preliminary, carries significant implications and raises the possibility of a trend towards increased opioid use among patients following their emergency department visit. These findings could potentially signal a need for heightened surveillance and targeted interventions to forestall the emergence of escalated opioid consumption subsequent to an emergency department visit.

Moreover, our study reveals a portrait of new opioid users, with a predominance of males aged 45-to-65 years. This noteworthy observation harmoniously aligns with the well-established trend of prescribing opioid analgesics for pain management within this particular demographic. The convergence of these demographic patterns and the burgeoning incidence of new opioid users further accentuates the complex interplay between age, gender, and opioid consumption patterns within the realm of emergency care. In addition, a study conducted by Serdarevic et al., 2017 [15] sheds illuminating insights on gender disparities in prescription opioid utilization. In contrast to our own dataset, women are more likely to use prescription opioids than men. Thus, the prudent course of action involves delving further into this domain through dedicated investigations.

### 4.2. Missing Data

Nonetheless, as we navigate the landscape of our findings, it is imperative to duly acknowledge certain limitations that have been brought to the forefront.

### 4.3. Subscription Compliance

Among our population, a subset of only 11 individuals (16.6%) exhibited a justified and compliant prescription of opioids, aligning with the guidelines set forth by the French National recommendations [10].

Standard recommendations were not followed when opioids were prescribed despite NRS scores falling below the threshold of 6; or when an extended-release formulation was opted for as the initial choice; or when the duration exceeded 14 days.

The prevalent opioid of choice was tramadol, with only 10 instances (16.7%) aligning with compliance guidelines.

These prescription patterns presented in Table 3 shed light on the intricate interplay between clinical decisions and guideline adherence. This exploration underscores the nuanced nature of opioid prescription practices within the context of emergency care and encourages us to reflect on the factors that contribute to such variations. It is important to recognize that while these findings highlight instances where prescriptions may deviate from recommended guidelines, they do not inherently indicate suboptimal practice; rather, they bring into focus the complexity of clinical decision making in the context of emergency care, where individual patient needs and clinical judgment play an essential role.

The recommendations on the use of opioids for acute pain were published close to the period of our study. Therefore, physicians’ knowledge was not up to date, and this could be the reason for deviation from the new guidelines. This study highlights the gap between the reality of prescription and the recommendations, especially in an emergency department (ED) with a massive flow of patients. In fact, in France, medical deserts are witnessed due a lack of general practitioners, which leads to the consultation of patients in the ED. This may lead to the long-term prescription of some treatments such as opioids, even when guidelines recommend a short-term prescription at ED as relayed by a general practitioner.

### 4.4. Tramadol Consumption

We delve into a crucial facet of new opioid users, primarily centered around tramadol as the principal agent. Tramadol’s categorization as a weak opioid (or level 2 according to the WHO classification) often belies its intricate nature, with its associated adverse effects and health controversies potentially underemphasized in the public consciousness. This classification as a “less potent” opioid probably explains this overrepresentation of tramadol. Faced with pain requiring opioid prescription, an emergency physician likely considers tramadol to be less risky for the patient, which is far from entirely true. The mean daily dose, quantified at 42 ± 19.5 mg morphine equivalent, sheds light on the daily opioid dosage regimen. While no strict maximum allowed equivalent morphine dose exists, Ref. [5] advocate against an increase of more than 50 mg of morphine equivalent per prescription per day. Moreover, unlike controlled substances like morphine or oxycodone, tramadol operates outside the purview of the same regulatory framework, possibly contributing to its less-stringently regulated prescription practices, which probably explain prescriptions exceeding 14 days.

Nevertheless, it remains imperative to underscore that tramadol is not devoid of inherent risks. Beyond its analgesic efficacy, tramadol carries a multifaceted profile replete with potential adverse effects. Its serotonergic component poses latent hazards, while its propensity to induce confusion adds an additional layer of concern. Furthermore, the heightened risk of falls underscores the pivotal necessity of vigilant monitoring and judicious utilization [16]. Additionally, insights from Langley et al., 2010 [17] offer valuable context. Their findings underscore that nausea, vomiting, dizziness, constipation, somnolence, and headache are common adverse events associated with various tramadol formulations. Notably, these events tend to be of mild-to-moderate severity and often manifest during initial treatment. The study also underscores potential differences in adverse event rates between long-acting and immediate-release tramadol formulations.

### 4.5. Strengths and Limitations

On the one hand, the first limitation is the relatively small sample size, which could potentially limit the generalization of the results. This study is monocentric, which, while confirming national results, could allow for center-specific effects. Additionally, this study adopts a strictly descriptive and retrospective approach, calling for caution in interpretation of the results. The findings, although informative, may require further prospective validation to confirm the observed trends.

Despite these limitations, the strengths of this study lie in its innovative nature, focusing on a precise observation of opioid prescriptions. This approach has enabled a comprehensive overview and the provision of supplementary information that contributes to optimizing opioid prescriptions while highlighting the significant prevalence of tramadol in adult emergency department prescriptions. Identifying and addressing these issues on the ground could potentially serve as an effective means to combat opioid misuse.

The intrinsic strengths of this study rest upon its relevance and originality in shedding light on opioid prescription patterns. The gathered information has the potential to guide targeted interventions aimed at optimizing prescription practices and enhancing patient safety within the context of emergency care. In the future, multicenter and prospective research could further substantiate these conclusions and confirm the trends highlighted by this exploratory study.

## 5. Conclusions

This study brings a significant contribution to our understanding of opioid prescription patterns among new users following an emergency department visit.

The revelation of tramadol’s predominant role among new opioid prescriptions underscores a complex dynamic. Despite its common categorization as a weak opioid, our analysis highlights that the prescribed quantities of tramadol align closely with those of morphine. This observation calls for a re-evaluation of tramadol’s perception and prompts consideration of its potential impact on patient health. The associated risks underscore the necessity for vigilant monitoring and careful management of its usage.

## Figures and Tables

**Figure 1 healthcare-12-01138-f001:**
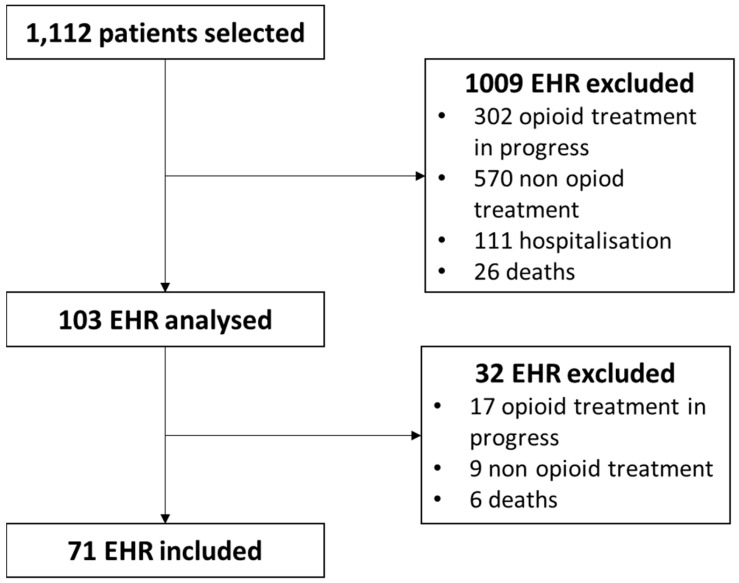
Patient selection.

**Table 1 healthcare-12-01138-t001:** Dose morphine equivalent used.

Drug	Adm.	DDD (mg)	DDD (mg) Morphine Equivalent	Equianalgesic Ratio
Morphine	PO	100	100	1
Oxycodone	PO	75	150	2
Codeine	PO	90/120	13.5/18	0.15
Opium	PO	Not available	Not available	0.10
Tramadol	PO	300	60	0.2

**Table 2 healthcare-12-01138-t002:** Characteristics of patients at admission in emergency department.

	Population (*n* = 71)
Age (years)	48.6 ± 21.4
Age group (years)	
[18–25]	12 (16.9)
[25–45]	21 (29.6)
[45–65]	23 (32.4)
≥65	15 (21.1)
Gender	
Female	31 (43.7)
Male	40 (56.3)
Numeric Rating Scale (NRS) at admission	6.9 ± 2.2
NRS Score	
Not available	13 (18.3)
≤3	3 (4.2)
4–5	16 (22.5)
≥6	39 (55.0)
Reason for hospitalization	
Traumatology	30 (42.3)
Rheumatology	13 (18.3)
Gastroenterology	11 (15.5)
Pneumology	7 (9.9)
Nephrology	5 (7.0)
Urology	1 (1.4)
ENT (Ear, Nose, and Throat)	1 (1.4)
Not specified	3 (4.2)

The results are presented as mean ± standard deviation or as *n* (%).

**Table 3 healthcare-12-01138-t003:** Compliance of opioid prescriptions of patients at emergency department discharge.

	Prescriptions	Compliant	Missing NRS	NRS < 6	Extended-Release Formulation	Missing Dosage and Frequency	Missing Duration	Duration (>14 Days)
Tramadol	60 (84.5)	10 (16.7)	11 (18.3)	17 (28.3)	8 (13.3)	3 (5.0)	12 (20.3)	11 (18.3)
Morphine	4 (5.6)	0 (0.0)	2 (50.0)	0 (0.0)	3 (75.0)	0 (0.0)	1 (25.0)	3 (75.0)
Oxycodone	3 (4.2)	0 (0.0)	0 (0.0)	0 (0.0)	1 (33.3)	1 (33.3)	3 (100)	0 (0.0)
Codeine	2 (2.85)	1 (50.0)	0 (0.0)	1 (50.0)	0 (0.0)	0 (0.0)	1 (50.0)	0 (0.0)
Opium	2 (2.85)	0 (0.0)	0 (0.0)	1 (50.0)	0 (0.0)	0 (0.0)	0 (0.0)	0 (0.0)
Total	71 (100)	11 (16.6)	13 (18.3)	20 (28.2)	12 (16.9)	4 (5.6)	17 (23.9)	14 (19.7)

The results are presented as *n* (%) or as median (IQR).

**Table 4 healthcare-12-01138-t004:** Opioid prescriptions of patients at discharge from the emergency department.

	*n* (%)	Cumulative Dose	In Morphine Equivalent	Daily Dose	In Morphine Equivalent
Tramadol	45 (88.2)	2517 ± 2068	503 ± 414	224± 93.0	44.7 ± 18.6
Morphine	3 (5.8)	520 ± 106	520 ± 106	33.3 ± 11.5	33.3 ± 11.5
Oxycodone	0 (0.0)	Na	Na	Na	Na
Codeine	1 (2.0)	840 ± Na	140 ± Na	120 ± Na	20 ± Na
Opium	2 (4.0)	405 ± 276	40.5 ± 27.6	45 ± 21.2	4.5 ± 2.12
Total	51 (100)	Na	480 ± 402	Na	42 ± 19.5

Na: Not applicable. The results are presented as mean ± standard deviation. The results are expressed in milligrams (mg).

## Data Availability

We used the institutional database of CHU Rouen called EdSAN (Health Data Warehouse of Normandy).

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
