# Peer review of "An Overview of Opioid Prescription Patterns among Non-Opioid Users Following Emergency Department Admission"

_healthcare, 2024, doi:10.3390/healthcare12111138_

Round 1
Reviewer 1 Report
Comments and Suggestions for Authors
1. The language and grammar needs a revision.
2. There are some minor typing errors
Abstract
- In the beginning of the Abstarct, you discuss about Tramadol. In the beginning of the abstract, you should shortly introduce the backround of your study. According to title, you study opioid prescription patterns. To my understanding, Tramadol prescription rate was a result and should be presented in the results and discussion section of the abstract.
Introduction
- Your “Introduction” starts about opioid use for chronic non-malignant pain. However, in your study, you study opioid prescriptions for acute pain and exclude patients with chronic pain and previous opioid use. Similarly, your seven first references are basically about opioids for chronic pain. Please, rewrite focusing in-line with your title. Also omit the less important references and replace them with more relevant ones, if possible.
Methods
1. I do not find Ethical committee approval or data protection information or information that they are not needed because of…
Results
1. The numbers in Flow Chart (Fig 1) are unnecessarily repeated in the text
2. Again, the results are unnecessarily repeated in the text and Table 2
3. Just curious: Can you really prescribe Opium in France?
Discussion
1. Your discussion could be condensed. For instance, your chapter 4.4 about Tramadol is too long and deviating too much from your title.
2. Conclusions: Too many superlatives. Although, your study is interesting clinically, it is not too innovative nor too significant. Your study is retrospective, one center study with quite a small number of patients as you have correctly stated in your “limtations”.
3. Conclusions: Here again, you are repeating the side effects of Tramadol unnecessarily. Please condense your conclusions to most relevant ones.
Tables and Figures
1. Table 4: You have quite large SDs. Perhaps Median [min, max] would be more informative?
Author Response
Dear reviewer,
thank you for your comments.
You will find the response document attached to this email.
Best regards,
the authors.

Reviewer 2 Report
Comments and Suggestions for Authors
This was an interesting read that would be of interest to Healthcare readers, but the information presented at this moment could be improved. In the light of public health, it is really important to explore different aspects of opioid addiction and to recognize healthcare professionals' role in it. However, I believe that this manuscript could benefit from some additional data or some sort of expansion as it feels pretty basic.
These are my comments to the respected authors.
Check the text for omitted and excess spaces (e.g. line 65, line 69, line 84, line 112…) and typos (e.g. Table 1, NRS score, “Not avaible”, line 149, “. Population”).
Introduction
Lines 34-36
Rephrase the first sentence – it lacks something. “…we took care of CNPC [with/using/by using/by...?]… a significant increase in the prescription of analgesics…”
Line 46
It would be advisable to put “Hider-Mlynarz et al.” instead of just a reference number at least as good practice.
Discussion
Subheading - Subscription compliance
Lines 182-189 are just a copy of the information presented in lines 131-137. There is no need to reiterate the same information here, in the discussion section. The authors can simply call on Table 3 and go on with the discussion.
In addition to the previously presented suggestions, I am of the opinion that the sample size is too small to extrapolate these data any further than the center where they were collected. The authors did mention this in the limitations, but I believe it limits this research tremendously. Could an additional center at least be added? Also, statistics are just basic frequencies and SDs. Could something be done to improve this?
Sincerely,
The reviewer
Author Response
Dear reviewer,
thank you for your comments.
You will find the response document attached to this email.
Best regards,

Reviewer 3 Report
Comments and Suggestions for Authors
The observational and retrospective single-center study on a small cohort of subjects attending the emergency department reports the prescription of opioids for analgesic purposes. The data confirm those reported in other larger studies, and represent an interest above all for the punctuality of the detection which confirms the trend towards over-prescription of tramadol in 2022 in an emergency department. The work, even if well organized, provides little new information. Tha major concerns are the lack of information about this prescriptive modality and the short follow-up. In particular, the scientific scope of communication would greatly increase if it were possible to verify the outcome of the 71 subjects enrolled after one year to see how long they used opioids, and whether they developed an opioid addiction. Furthermore, in discussing the results, it would be appropriate for the authors to also discuss the reasons which, in their opinion, underlie the prescription of the drug in ways that do not comply with the indications of the Guidelines of the French National Authority for Health and what the real or presumed advantages would be in the prescription of tramadol versus codeine.
Author Response

(The authors gave the same response as above.)

Reviewer 4 Report
Comments and Suggestions for Authors
it could be interesting to carry out a multicenter national and then international European study. Restrictions have recently been added to prescriptions for tramadol as well. It could be useful to perform an analysis of all opioids prescribed and trace them to specialties (orthopaedics, anesthesiology, rheumatology, oncology)
The article is interesting and timely precisely because of the incipient and growing attention on opioid drugs. It might be interesting to understand which category of doctors liberally prescribe drugs. The methodology seems adequate although it would be appropriate to extend it to other centers. The conclusions are consistent with the evidence and arguments presented.
Author Response

(The authors gave the same response as above.)

Round 2
Reviewer 3 Report
Comments and Suggestions for Authors
The new version presented collects the observations made and, while not modifying the general evaluation of the study, better outlines the limitations and problems inherent in the prescription of opioids in an emergency department in France. In the current version the article is publishable.